# Recent Developments in Magnetic Hyperthermia Therapy (MHT) and Magnetic Particle Imaging (MPI) in the Brain Tumor Field: A Scoping Review and Meta-Analysis

**DOI:** 10.3390/mi15050559

**Published:** 2024-04-24

**Authors:** Frederika Rentzeperis, Daniel Rivera, Jack Y. Zhang, Cole Brown, Tirone Young, Benjamin Rodriguez, Alexander Schupper, Gabrielle Price, Jack Gomberg, Tyree Williams, Alexandros Bouras, Constantinos Hadjipanayis

**Affiliations:** 1Department of Medical Education, Icahn School of Medicine at Mount Sinai, New York, NY 10029, USA; frederika.rentzeperis@icahn.mssm.edu (F.R.); daniel.rivera@icahn.mssm.edu (D.R.); jack.zhang@icahn.mssm.edu (J.Y.Z.); cole.brown@icahn.mssm.edu (C.B.); tirone.young@icahn.mssm.edu (T.Y.); gabrielle.price@icahn.mssm.edu (G.P.); jack.gomberg@icahn.mssm.edu (J.G.); 2Sinai BioDesign, Department of Neurosurgery, Mount Sinai, New York, NY 10029, USA; willit13@rpi.edu; 3Department of Neurological Surgery, Mount Sinai Hospital, New York, NY 10029, USA; alexander.schupper@mountsinai.org; 4Department of Biomedical Engineering, Rensselaer Polytechnic Institute, Troy, NY 12180, USA; 5Brain Tumor Nanotechnology Laboratory, UPMC Hillman Cancer Center, Pittsburgh, PA 15232, USA; bourasa@upmc.edu; 6Center for Image-Guided Neurosurgery, Department of Neurological Surgery, University of Pittsburgh School of Medicine, Pittsburgh, PA 15213, USA

**Keywords:** magnetic hyperthermia therapy, magnetic particle imaging, nanoparticles, glioma, glioblastoma, astrocytoma, hyperthermia

## Abstract

Magnetic hyperthermia therapy (MHT) is a promising treatment modality for brain tumors using magnetic nanoparticles (MNPs) locally delivered to the tumor and activated with an external alternating magnetic field (AMF) to generate antitumor effects through localized heating. Magnetic particle imaging (MPI) is an emerging technology offering strong signal-to-noise for nanoparticle localization. A scoping review was performed by systematically querying Pubmed, Scopus, and Embase. In total, 251 articles were returned, 12 included. Articles were analyzed for nanoparticle type used, MHT parameters, and MPI applications. Preliminary results show that MHT is an exciting treatment modality with unique advantages over current heat-based therapies for brain cancer. Effective application relies on the further development of unique magnetic nanoparticle constructs and imaging modalities, such as MPI, that can enable real-time MNP imaging for improved therapeutic outcomes.

## 1. Introduction

Gliomas are the most common primary central nervous system (CNS) malignancy and are thought to derive from neuroglial stem or progenitor cells of the CNS [1]. Gliomas are classified by the World Health Organization (WHO) and range from low-grade gliomas (LGG, WHO grade I-II) to high-grade gliomas (HGG, WHO grade III-IV), with HGGs being associated with an extremely poor prognosis [2,3]. Nearly half of all gliomas are classified as glioblastomas (GBM), a rapidly growing HGG notable for its diffuse infiltrative growth pattern and extensive cellular and molecular heterogeneity [4,5,6]. Although LGGs may be curable with surgical resection, GBM remains difficult to treat despite an aggressive standard of care consisting of maximal safe tumor surgical resection and concomitant temozolomide (TMZ) chemotherapy (CT) and radiation therapy (RT) [7,8,9,10,11,12,13]. Recent studies have shown that a highly heterogeneous sub-population of therapy-resistant cells, referred to as GBM stem-like cells (GSCs), often reside in the infiltrative margin of the tumor and evade surgical resection. GSCs have been identified as key mediators of therapy resistance, leading to invariable lethal local recurrence [14,15,16]. Despite recent advances in intraoperative visualization and decades of research into novel therapies, the median survival for patients with GBM remains only 15–18 months [17,18].

One therapeutic approach with significant potential for the treatment of GBM is hyperthermia therapy (HT), where the temperature is increased above the baseline body temperature. When tumors are heated between 40–45 °C, numerous changes occur that are toxic to both tumor vasculature and the cancer cells themselves [19,20,21,22]. Magnetic nanoparticles (MNPs) are a useful tool for performing a highly localized form of HT, known as magnetic hyperthermia therapy (MHT). After direct intratumoral deposition or the systemic delivery of the MNPs, a safe external alternating magnetic field (AMF) is applied to heat the MNPs [23]. MHT offers many unique advantages over other common heat-based therapies for GBM. Namely, the penetration depth of the AMF is greater than that of the other activation modalities used in HT, such as light or acoustic waves. This allows for the heating of deeply seated tumors without having to remove bone or perform skin incisions once the MNPs are intracranially deposited by stereotactic injection or by convection enhanced delivery (CED) [24]. Additionally, adjusting the AMF amplitude can allow for the precise control and regulation of heating [25]. Moreover, MNPs remain around the injection site for weeks to months, allowing for multiple MHT sessions to be performed after a single delivery of MNPs [26,27,28]. MHT has also been shown to radiosensitize GSCs, and it may produce a more uniform temperature distribution across the target lesion when compared to other thermal therapies such as laser interstitial thermal therapy (LITT) [29,30,31]. MHT is a locally confined, remotely controllable, and easily reproducible form of HT with the potential to be highly relevant to the future treatment of GBM.

A crucial element in the application of effective MHT is the accurate localization of the nanoparticles prior to (and, ideally, during) the application of an AMF. Magnetic nanoparticles have been utilized as contrast agents for magnetic resonance imaging (MRI), owing to their ability to shorten MR relaxation times, leading to potential roles as T1, T2, or dual contrast agents, depending on their formulation [32]. Despite the previous applications of MRI in imaging MNPs, this imaging modality is limited by its low penetration depth, poor resolution, and magnetic artifact. Magnetic particle imaging (MPI) serves as an alternative imaging modality for MNPs with theoretically unlimited penetration depth and no tissue background noise [33]. As originally described by Gleich and Weizenecker in 2005, MPI leverages the nonlinear magnetizability of the injected MNPs—in an AMF comprised of harmonic frequencies and a drive frequency, the MNPs’ magnetization will saturate above a given field strength [34]. An additional magnetic field that is zero at the center (the field free point) and increasing toward the edges is simultaneously delivered, so that all magnetic material outside the field free point (FFP) will be saturated and the MNPs at the FFP will exhibit a harmonic signal [34]. A map of the nanoparticles’ localization can be generated by navigating the FFP throughout the volume of the region of interest and co-registering this image with other structural imaging modalities like MRI.

Several of the novel magnetic nanoparticles (MNPs) to be discussed in this review are biodegradable and, thus, will decrease in concentration over time; additionally, non-biodegradable MNPs will eventually be cleared from the tissue to which they are delivered. It is, thus, additionally important to quantify the amount of magnetic material present in the tumors prior to the application of an AMF to determine if additional injections of MNPs are necessary. While the quantification of iron levels in the liver of patients with hereditary hemochromatosis has been carried out using MRI, the use of MRI for iron quantification in MHT is limited due to the signal saturation resulting from the high MNP concentrations required [35,36]. The presence of fat or water in the tissue may also lead to oscillating signals at a given voxel, thereby potentially further obscuring signal; moreover, the required scan times are long, so there is an associated high cost for the scans and risk of motion artifacts [35]. This serves as further motivation for the exploration of MPI in the context of MHT for brain tumors.

## 2. Methods

This scoping review was reported according to the Preferred Reporting Items for Systematic Reviews and Meta-Analyses extension for Scoping Reviews (PRISMA-ScR). This review on MNPs for CNS cancers is divided into two points of interest, (1) MHT and (2) MPI, to visualize the spatial organization of the MNPs. This review was conducted systematically through the leading journals and search engines in the field, including Pubmed, Scopus, and Embase using the search terms (“magnetic hyperthermia” OR “magnetic particle imaging”) AND (“brain cancer” OR “brain tumor” OR glioblastoma OR astrocytoma OR glioma).

Duplicate articles were removed by Covidence. Two authors independently reviewed the articles for inclusion in an abstract/title review followed by a full-text review. At the abstract/title stage, articles were excluded if they were not original research articles (reviews, opinions, book chapters), were conference abstracts, were not available in English, or were unavailable due to a paywall. At the full text stage, articles were included if they pertained to CNS cancers, animal work performed in orthotopic brain tumor models (subcutaneous tumor xenografts were excluded) and focused on MHT or MPI (Figure 1).

## 3. Results

### 3.1. Magnetic Hyperthermia

Of the 12 papers selected for the review article by two independent reviewers, 9 pertained to the application of MHT in brain cancers and included an animal study as an experimental component. The following section will summarize the experiments carried out with the various MNP types used, their physical properties, and their efficacy in the treatment of brain tumors both in cell culture and animal models. A summary of the studies included is tabulated below in Table 1.

#### 3.1.1. Nanoparticle Characteristics

The following are descriptions of the experimental conditions and novel characteristics of the various MNPs tested in GBM/glioma animal models, as illustrated in Figure 2.

Magnetosomes [1]

Alphandéry et al. [37] evaluated a biodegradable nanoparticle generated by AMB-1 *Magnetospirillum magnetotacticum*—a species of magnetotactic bacteria with unique organelles called magnetosomes that encapsulate ferromagnetic crystals and allow their orientation to, and migration along, geomagnetic field lines [46]. These MNPs are of particular interest as they are biodegradable and, thus, will not accumulate for a prolonged period of time in the target tissue after their administration. Additionally, they preferentially arrange in chains, thereby preventing their aggregation, which is beneficial for MHT as it reduces the risk of embolism and improves the uniformity of the heat output. Furthermore, the magnetosome organelles contain endotoxins in their lipopolysaccharide core, resulting in therapeutically advantageous cytotoxicity and immune recruitment in animal studies.

Seventy mice were intracranially inoculated with GBM cells via stereotactic injection, after which the tumors were permitted to grow for 8 days. Experiments were conducted with commercially available IONPs purchased from Micromod. After day 8 post-tumor cells implantation (PTI), the mice were assigned into seven treatment groups, including four controls that received the following: glucose, glucose + AMF, magnetosomes without AMF, and IONPs without AMF. In two experimental groups, animals were stereotactically injected with 13 μg/mL magnetosomes + 15 AMF sessions with different tumor volumes (3 mm^3^ and 25 mm^3^). In one experimental group, animals with tumor volumes 3 mm^3^ received IONPs + 12 AMF sessions.

This was the first of three papers by Alphandéry et al. [37] focused on the application of magnetosomes in GBM animal models; this paper reported their efficacy and established them as a viable alternative to commercially available IONPs. This paper additionally laid the groundwork for their future publications on the biodegradability of the magnetosomes and the use of a Poly-L-Lysine (PLL) coat.

Despite the promising results, it is important to recognize that the specific interactions between magnetosomes and various tumor microenvironments remain poorly understood, potentially limiting the generalizability of said findings across different GBM presentations, necessitating further research to optimize therapeutic efficacy.

PLL Coated Magnetosomes [2]

In their 2017 paper, Alphandéry et al. [38] purified magnetosomes to remove endotoxins and organic material and coated them with PLL, henceforth called “M-PLL”. In preliminary experiments, uncoated magnetosomes aggregated, increasing the risk of embolism, so the magnetosomes were coated with PLL to prevent aggregation and improve their heating capacity. Experiments were conducted with the same commercially available IONPs as in the first paper.

Mice were implanted with U87-Luc GBM tumor cells. After 5 days, mice received intratumoral injections of glucose, 500 μg IONP, or 500 μg M-PLL, followed by either no further treatment or 23–27 magnetic sessions (on D5–D63 PTI) with an AMF of 27 mT and 202 kHz for 30 min. A repeat treatment with 200 μg of nanoparticles was administered on D47 PTI; in mice, this resulted in tumor regrowth. In the control animal groups that received glucose with or without AMF, exponential tumor growth was observed until D40–54 PTI, at which point the mice were euthanized. Mice that received IONPs with no AMF had similar outcomes. Contrastingly, mice administered M-PLL without AMF survived, on average, until day 111 PTI, suggesting the cytotoxicity of M-PLL and some degree of anti-tumor activity. Both the IONPs and M-PLL groups showed enhanced anti-tumor efficacy and prolonged overall survival with AMF application. In the IONPs + AMF animal group, 2 mice had complete tumor regression while the remaining 7 exhibited delayed tumor growth relative to controls. In the M-PLL + AMF group, 5 mice had continuous tumor regression and 4 experienced tumor recurrence but achieved complete regression upon the second treatment, resulting in 100% survival by D350 PTI.

The authors suggest that the improved outcomes in the M-PLL group were likely due to stronger and longer-lasting heating by M-PLL. Indirectly, M-PLL showed 63% apoptotic death compared to 8% with IONPs, indicating a “thermal bystander effect” by which tumor cells respond to the apoptosis of their neighbors. Polynuclear neutrophils (PNN) were observed 6 h after M-PLL administration and were likely involved in the cytotoxicity of the surrounding tumor cells. Their recruitment was not, however, due to the presence of endotoxins as these were confirmed to be at the same low level as in IONPs (which did not show PNN recruitment).

The long-term biocompatibility and potential immunological responses elicited by PLL-coated magnetosomes in diverse biological systems warrant further investigation, underscoring the need for comprehensive in vivo studies to fully assess their safety and therapeutic viability.

Magnetosome Chains [3]

In their most recent 2019 article, Alphandéry et al. [39] evaluated their biodegradable magnetosomes for complete tumor eradication in GBM. This study served as initial proof that MNPs could be sufficiently small in size in order to be taken up and degraded by the host tissue while remaining effective in yielding complete tumor regression in 50% of animals. U87-Luc GBM cells were intracranially inoculated into mice, followed one week later by the stereotactic administration of either vehicle or 2 μL of 40 mg of magnetosomes. Subsequently, the mice were exposed to either no magnetic field or an AMF at 27 mT and 198 kHz for varied numbers of sessions; namely, either 3 or 12 magnetic sessions after vehicle administration, or 15 magnetic session after nanoparticle administration. Intratumoral temperature was recorded during the sessions, and bioluminescence imaging (BLI) was performed after each session. Due to the biodegradability of the magnetosomes, tumor temperature no longer increased after the 5th AMF session, but the anti-tumor effects persisted.

The authors proposed that the continued efficacy may be attributed to the immune response to magnetosome exotoxins, toxicity from intracellular heating/iron release due to magnetosome internalization by tumor cells, or an apoptotic mechanism induced by the heating.

While the persistent anti-tumor effects post-degradation of magnetosomes highlight their potential, the inconsistency in the response among individual tumors underscores the complexity of translating these outcomes to a clinical setting, where patient-specific factors may significantly influence therapeutic effectiveness and safety.

Chitosan Coated Fe_3_O_4_ [4]

Most preclinical studies of MNPs evaluated clearance at concentrations well below the doses necessary for MHT, which resulted in the recall of several nanoparticles due to high immunogenicity, teratogenicity, and excessively long clearance times during clinical administration. To avoid the risk of recall upon clinical administration, this study aimed both to establish a research paradigm for future MHT studies to evaluate the toxicity, clearance, and efficacy of MNPs, and presented a complete profile for a novel nanoparticle.

They first generated Fe_3_O_4_ nanoparticles electrochemically and then modified their surfaces with a chitosan polymer coating. The chitosan coating is a lipophilic cationic layer that improves stability in a colloid, biocompatibility, heating efficiency, and leads to faster magnetic relaxation. To evaluate the nanoparticles’ effects in animal models, male Wistar rats were treated with a mixture of immunosuppressive agents to enable the injection of human C6 GBM cells. After a week of immunosuppression, 5 × 10^6^ C6 cells were implanted into the flanks of the rats and allowed to grow for 12–14 days until they were 300–500 mm^3^ in diameter. At that time, nanoparticles were delivered with 2 μg/mm^3^ of tumor, directly to the tumors by injection in three locations (at 3, 6, and 9 o’clock points). The rats were assigned into 3 groups; 6 rats received vehicle + AMF, 6 received nanoparticles without AMF, and 6 received nanoparticles + AMF at 335 kHz for 20 min. The rats received the same treatments a week later, but with 1.5 μg/mm^3^ of tumor.

The authors reported that neither component of the MHT had an effect on tumor size in isolation—AMF without nanoparticles and nanoparticles without AMF resulted in the same tumor growth curves as the no-treatment group. With combined chitosan nanoparticles + AMF, nanoparticle temperatures increased from 32–42 °C within 400 s and remained at 42 for the duration of the magnetic session; tumor growth in these rats was completely inhibited by day 32 PTI. This study demonstrated faster tumor clearance with fewer and shorter AMF sessions than previous studies using chitosan-coated Fe_3_O_4_ nanoparticles. The primary mechanism of cell death in response to the MHT was apoptosis, which can lead to less inflammation than a necrotic mechanism.

Over the 5 months following MHT, the authors evaluated MNP levels in blood, feces, urine, and organs. In the first month after their injection, high levels of nanoparticles were detected in the urine, after which their concentration declined. Similarly, MPNs were detected in the feces at high levels for the first 3 months, after which their concentrations fell off. As for the organs, the liver exhibited elevated iron levels for the first 3 months, but other major organs like kidney, lung, and heart did not show significant MNP accumulation. All parameters were kept within previously determined clinical safety limits, including the AMF, temperature (<45), and concentration of iron (<22.4 μg/mm^3^ of tumor).

While the study underscores the efficacy and safety of chitosan-coated Fe_3_O_4_ nanoparticles for magnetic hyperthermia therapy, it primarily focuses on subcutaneous tumor models, which may not fully replicate the intricate microenvironment and therapeutic challenges posed by intracranial tumors, suggesting a need for further research in models that closely mimic human brain tumor conditions.

Disk-Shaped Permalloy Magnetic Particles—Mechanical Tumor Disruption [5]

Instead of applying hyperthermia to tumors via nanoparticles, Cheng et al. [41] leveraged disk-shaped particles and a rotating magnetic field to exact a mechanical force and shear tumor cells directly. They generated disc-shaped nanoparticles composed of a sandwich of 5 nm Au, 60 nm permalloy (Ni_80_Fe_20_), and 5 nm Au. They carried out two experiments, one in which the magnetic discs were incubated with U87-Luc GBM cells for one day prior to the injection of the nanoparticle-infiltrated glioma cells (survival study), and another in which tumor cells were injected into the mice and nanoparticles were delivered via the same injection site 3 days post-tumor induction (histology apoptosis study). In both cases, 10^5^ cells were intracranially implanted along with 5 × 10^6^ nanoparticles.

A 20 Hz, 1 T rotating magnetic field was administered for 7 days beginning 4 days PTI. No significant changes in temperature were detected, but the treatment group showed significant intratumoral apoptosis without affecting the surrounding normal brain tissue. The control group with nanoparticles only exhibited continued tumor growth while those that were exposed to the rotating magnetic field with nanoparticles showed tumor regression by day 7 post-tumor implantation. By day 28, 40% of the animals had no tumor signal on BLI; the treatment group also exhibited prolonged survival. While the control group mice showed a median survival time of 56 days, the treatment group exhibited a median survival of 63 days. After treatment, the nanoparticles were found intratumorally but not in the surrounding healthy brain tissue or in any other organ (liver, spleen, heart, lungs, large intestine, kidney, bladder, and testes), indicating that the particles are difficult to be cleared from the brain, but also that they do not accumulate elsewhere.

Despite the innovative approach of using disk-shaped permalloy magnetic particles for mechanical tumor disruption, the method’s reliance on specific nanoparticle shapes and the need for a rotating magnetic field may limit its adaptability and scalability across different tumor types and clinical settings, highlighting the necessity for further exploration of the practical implementation and long-term outcomes of this technique.

Gallic Acid-Coated Magnetic Nanoclovers [6]

Liu et al. [42] aimed to circumvent the two major issues in current approaches to MHT—inefficient nanoparticle heating and poor blood–brain barrier (BBB) penetration. They generated cobalt-doped nanoparticles under various reaction conditions and found that their “nanoclovers” with diameters of 20.7 nm were the most efficient at raising intratumoral temperatures, attaining temperatures in the range of 45.6–50.2 °C. Without a coating, the nanoclovers aggregated and displayed reduced heating efficiency, so they tested several polyphenol coatings and landed on gallic acid to maximize stability and dispersity. Further, gallic acid is known to bind to vascular endothelial growth factor 2 (VEGFR2), which is expressed in tumor vasculature, including GL261 gliomas, but not in healthy vessels, thus enabling tumor vessel targeting and the systemic administration of the nanoclovers.

To deliver the nanoclovers, they performed a tail vein injection of MNPs in mice with GL261 gliomas expressing GFP. Subsequent to the delivery of nanoclovers, they found that 13.5% of injected nanoclovers localized to the tumors, which was 20.1× the localization to the healthy brain. At 2 h after injection, the nanoparticles were largely localized to the tumor vessels, and progressively extravasated into the tumor tissue over the following 24 h. They, thus, chose to perform MHT at 12 h post-nanoparticle injection to maximize damage to both tumor vessels and the tumor parenchyma.

After treatment with MNP and AMF, they administered Evans blue dye to evaluate vessel leakiness and paclitaxel (PTX), a chemotherapy drug with minimal BBB penetration, to evaluate whether MHT could enhance chemotherapy delivery to tumors by MHT-induced BBB disruption. A 1.5-fold increase in the PTX accumulation leakage of dye into tumors was found, indicating that there is disruption of the BBB by MHT and improved drug delivery. As for the survival studies, the nanoparticles + AMF did not completely eliminate the tumors but did prolong survival of mice by 52 days compared to controls. They additionally evaluated the intracranial implantation of nanoparticles with CED and observed a similar outcome as with intravenous nanoparticle delivery. Administration of PTX without MHT led to no survival benefit over controls but PTX administration followed by MHT led to a reduction in tumor size relative to MHT alone, suggesting that future studies of combined chemotherapy + MHT would be valuable. This study was relatively short-term and only a single session of AMF was applied; additional studies should be carried out as the residual tumor would likely regrow and, with substantial disruption to the BBB, there is a risk for metastasis.

While the innovative use of gallic acid-coated magnetic nanoclovers shows promise for overcoming BBB penetration and enhancing MHT efficiency, the potential for unintended BBB disruption raises concerns about systemic toxicity and the risk of enabling metastatic spread, emphasizing the critical need for further studies to balance therapeutic efficacy with safety.

SPIONa [7]

Rego et al. (2019) [43] highlighted the significant promise of MHT for primary brain cancer therapy. However, they noted that there are still significant gaps in the literature regarding the evaluation of each nanoparticle type. In particular, most studies on SPIONs used for GBM therapy have been conducted on tumors engrafted in the flanks of animals, which have a significantly different microenvironment than intracranial tumors. The constrained volume of the skull and mass effects from intracranial tumors, as well as the differences in physiology including local metabolism, perfusion, and the coefficients of the heat transfer of the neighboring tissue, directly influence the heating capacity of MNPs. Therefore, the authors selected aminosilane-coated superparamagnetic iron oxide nanoparticles (SPIONa)—a MNP type with promising results from previous studies—and aimed to conduct a more comprehensive evaluation of its physical properties and the optimal conditions for the applied magnetic field.

Previous studies on SPIONa showed that they have a high saturation magnetization value, form stable deposits throughout tumors and that they did not damage surrounding cortical cells in animal models during MHT. Rego et al. [43] carried out in vitro viability assays on C6 cells transduced with luciferase and subsequently incubated with 600 mg SPION/mL for 18 h. C6 cells with and without IONPs were then either allowed to sit or exposed to an AMF of 874 kHz at 200 Gauss for 40 min, after which they were evaluated for viability by bioluminescence (BLM). There was no significant difference in the survival of tumor cells with AMF or nanoparticles alone; cell viability was reduced 52% when cells were treated with MHT.

They further conducted animal studies on 2-week-old male Wistar rats by stereotactically injecting 10^6^/10 μL C6 glioma cells in the right frontal cortex. Twenty-one days PTI, control values for BLM were obtained for all the rats prior to the administration of MHT on day 22. They delivered a total of 0.5 mg of SPIONa into four equidistant points about the tumor centroid, which was the lowest mass of SPIONa administered in comparable studies (0.5–3 mg), to minimize toxicity and obtain a lower bound on the nanoparticles’ efficacy. Twenty minutes later, they delivered 874 kHz in a field of 200 Gauss until a tumor temperature of 42 °C was obtained, as measured by optical fiber temperature probe, after which the field strength was modulated at the same frequency to sustain a temp of 42 °C for 40 min. A 32.8% reduction in tumor mass was measured on BLM after MHT.

SPIONa [8]

Rego et al. (2020) [44] expanded upon their 2019 study by demonstrating MHT efficacy in cell culture and animal models. Their study consisted of an evaluation of the heating potential of SPIONa nanoparticles, followed by cell culture and animal experiments in rats. After measuring the heating time needed for SPIONa to achieve a therapeutic temperature (43 °C), the authors tested different oscillation frequencies and magnetic field strengths before deciding on two optimal combinations. These parameters—557 kHz, 300 Gauss, and 309 kHz, 300 Gauss—were then applied to the subsequent cell culture and animal studies.

Utilizing the C6 glioma cell line for the in vitro experiments, the authors first studied SPIONa internalization. Following a 12 h incubation of 100 and 200 μgFe/mL of SPIONa in culture, the cells were washed, fixed, and stained with prussian blue and nuclear fast red. The authors reported that both concentration conditions showed SPINOa internalization, with the higher 200 μgFe/mL condition demonstrating greater internalization on microscopy. To evaluate MHT efficacy of the two aforementioned sets of parameters, the authors conducted MHT in three stages and evaluated therapeutic efficiency through C6 cell BLI signal intensity. MHT was applied for 30 min in three sessions on days 0, 3, and 6 post-tumor induction, and BLI measurements were conducted on days 2, 5, and 8, respectively (48 h after each MHT). Results from the first two MHT sessions showed that cell viability was more significantly reduced in the higher (557 kHz), as opposed to the lower (309 kHz), AMF conditions after the first MHT session (20.02% compared to 40.03%, respectively) as well as after the second (12.49% compared to 25.02%, respectively) MHT session. Interestingly, the authors noted that viability was inversely proportional to frequency following the third MHT session and speculated that this could be related to the development of heat–stress resistance.

Moreover, the authors intracranially implanted C6 glioma cells into male Wistar rats and evaluated the efficacy of the same two sets of MHT parameters used in the in vitro experiments. In total, 3 MHT sessions occurred on day 14, 17, and 21PTI, and BLI signal was recorded one day prior to each MHT session, and after 2 and 12 days following the third session, in order to monitor tumor growth. Additionally, positron emission tomography (PET) was used before and after each MHT session to monitor glucose uptake by tumor cells, and locomotor assessment was conducted throughout. The authors reported that BLI signal reduction was directly proportional to the number of MHT sessions—29.7%, 61.4%, and 94.9% for one, two, and three sessions, respectively—and that the decrease in tumor growth compared to the baseline was statistically significant. They also reported that animals receiving three sessions of MHT showed an absence of tumor relapse at day 32 PTI and sustained decrease in BLI signal at late evaluation, findings that are supported by PET. Finally, they reported that while animals receiving only one or two MHT sessions failed to demonstrate significant symptomatic improvement, those receiving three MHT sessions demonstrated significant improvement in horizontal movements by day 19 PTI compared to other tumor groups; this persisted until study conclusion at day 32 PTI.

Although SPIONa demonstrates potential for MHT in brain cancer therapy, the study’s focus on a single animal model and the limited diversity of experimental conditions highlights the necessity for extensive multi-model research to fully understand the nuanced effects and optimal usage of SPIONa across a broader spectrum of brain tumor types and microenvironments. Additionally, SPIONs are well-known to be uptaken by the reticuloendothelial system, so accumulation in potentially unwanted organs such as the liver and spleen is a possibility.

Zinc- and Cobalt-Doped Cubic Iron Oxide Nanoparticles [9]

The work of Wu et al. [45] introduced zinc- and cobalt-doped cubic iron oxide nanoclusters (ZnCoFe NCs), innovatively synthesized for high-grade glioma treatment. These nanoparticles underwent a precise thermal decomposition method, where Zn(acac)_2_·H_2_O and Co(acac)_2_ were mixed with Fe(acac)_3_ in specific ratios. This methodical doping is key to manipulating their magnetization and anisotropy energy, directly influencing their heating efficiency in MHT. To ascertain the structural integrity and uniformity in the composition, the authors utilized X-ray diffraction for crystallinity analysis, scanning electron microscopy-energy dispersive X-ray spectrometry for elemental distribution mapping, and X-ray photoelectron spectroscopy to confirm the chemical state and binding energy of the doped elements.

The magnetic properties of ZnCoFe NCs were rigorously examined using hysteresis loops and physical property measurement system analysis. The ferrimagnetic nature of these nanoparticles was evident, with Zn_0.4_Co_0.4_Fe_2.2_O_4_ showing an exceptional specific absorption rate (SAR) of 3890 W/g metal. This high SAR value underscores the capacity of these NPs for effective heating under an alternating current magnetic field, making them suitable for hyperthermia applications. In their animal GBM mouse model, a post-blood–brain-tumor barrier (BBTB) modulation experiment was performed, during which these nanoparticles were tagged with near-infrared dye IR780 or coumarin 6, and they demonstrated an enhanced accumulation in the tumor region. This finding highlights their potential effectiveness in crossing the BBTB and their targeted delivery efficiency.

Evaluating their therapeutic efficacy, the ZnCoFe NCs, particularly when used in conjunction with BBTB modulation, showed substantial tumor suppression in GL-261 and U87 GBM mouse models. The addition of the heat shock protein inhibitor, VER-155008, to the treatment significantly enhanced the hyperthermia effects, prolonging mouse survival and providing a possible adjuvant to AMF—which was not tested in this study. Finally, the biocompatibility of ZnCoFe NCs was thoroughly assessed. Intravenous NP administration showed minimal cobalt ion leaching, and no significant changes in animal body weight or negative impact on major organs were noted.

While the introduction of zinc- and cobalt-doped cubic iron oxide nanoparticles represents a significant advancement in the field of MHT, the comprehensive evaluation of their long-term biocompatibility and potential systemic effects following their use remains crucial. Further studies are needed to assess the full impact of these nanoparticles on the overall health and function of treated organisms, especially considering the enhanced permeability they may induce in the blood–brain barrier.

#### 3.1.2. Tabulated Experimental Parameters

In Table 2, we present various parameters for the nanoparticles evaluated in the nine papers of interest, including the following: mean particle size, coercivity (H_C), remnant-to-saturating magnetization ratio (M_r/M_S), mass, volume, specific absorption rate (SAR, aka specific loss power), and change in temperature (Δt).

In Table 3, we describe the various parameters pertaining to the animal experiments, including the number of animals, age, sex, strain, MNP injection day PTI, MNP delivery method, and survival of the animals. MS = MHT Sessions.

### 3.2. Magnetic Particle Imaging

Accurate localization of nanoparticles is crucial to the delivery of MHT. At present, however, research on intracranial imaging of MNPs has been incredibly limited. The following section describes the recent advances in MPI as applied to intracranial tumors.

T-Cell Immunotherapy Tracking

Adoptive cellular therapy (ACT) is an immunotherapeutic technique through which a patient’s tumor-infiltrating T-cells are collected, grown ex vivo, and then re-infused into the patient to augment their immunologic antitumor response. This therapeutic approach has been applied across tumor types and has shown particular promise in the context of metastatic melanoma, exhibiting a 40% clinical response rate in brain metastases. Despite its promise in metastatic disease, the approach has been largely unsuccessful in primary brain cancers, including GBM, due to reduced trafficking and the retention of the T-cells. Nevertheless, successful intracranial trafficking in the context of melanoma indicates potential promise for the technique for other tumors in the brain.

In their 2021 paper, Rivera-Rodrigeuz et al. aimed to leverage MPI to track T-cells during ACT in the mouse models of brain cancer. Previously, attempts to track lymphocytes with PET, SPECT, and MRI with tracers were unsuccessful due to poor penetration depth, resolution, and artifacts, despite the approach’s promise of the theoretically infinite penetration depth of MPI and high resolution. T-cells were labeled ex vivo with ferucarbotran, a magnetic tracer, and were then administered to the mice intravenously or intraventricularly. A linear relationship was established between T-cells and iron detection with MPI. The labeling had no impact on cell viability or functionality in the T-cells.

3D Printed Mouse Phantom for MPI

To aid in the optimization of future MPI studies, Sarna et al. [47] engineered an anatomically relevant mouse phantom with several hollow organ-representative cavities generated from the Digimouse atlas. The brain cavity contained a tumor simulacron, as well as two capillary tubes used for SPION delivery. Following the 3D printing of the model, an iron mass of approximately 10 μgFe was placed into the brain tumor compartment to serve as a constant tracer. This fixed mass was selected to match the experimental conditions used by Rivera-Rodrigeuz et al., as discussed previously. Additionally, a dilution series of tracer iron mass was placed within the tumor region itself to enable signal comparison between this region of interest and that of the fixed iron mass. MPI scans were then acquired in both 2D and 3D, as well as in both high sensitivity (HS) and high sensitivity/high resolution (HSHR) modes.

For the brain tumor model, the authors reported excellent agreement between calculated and known tracer mass across all the SPION dilutions used and MPI modes tested. These findings suggest that such a phantom model has utility in assessing the accuracy of intracranial MPI, prior to initiating time-intensive and costly experiments in animal glioma models.

Human MPI scanners

Upscaling MPI scanners for human use is a current challenge which limits the ability to assess this technology in combination with MHT in clinical trials. The first ever human-sized MPI scanner designed for brain application was described by Graeser et al. in 2019 and marked a large milestone in the clinical application of MPI [48]. Limitations of this device included relatively low spatial resolution and sensitivity. Increasing sensitivity and resolution often requires using stronger magnetic field gradients. However, as field gradient strength increases, so too does the risk of unwanted side effects, such as peripheral nerve stimulation. The current challenge facing researchers is designing a scanner that strikes a balance between optimal imaging capabilities and patient safety.

Multimodal Imaging and Future Applications of MPI

Multimodal imaging refers to combining distinct imaging modalities into a single platform to image a single subject. In their 2019 paper, Song et al. [49] described a novel multimodal imaging platform (MMPF NP) that combined MPI, MRI, photoacoustic imaging, and fluorescent imaging, which they used to image xenografts in a live mouse model. Orthotopic GBM xenografts were generated through the stereotactic implantation of luciferase-transfected human glioma cells (U87-Luc), and the mice were injected intravenously with the MMPF NPs prior to imaging.

Results showed not only clear visualization of the GBM tumor bulk through intact skull, but also that MMPF NP injection was associated with a 17.1-fold MPI signal enhancement, significantly outperforming conventional MRI comparisons. Furthermore, upon the dissection of the brain for ex vivo validation, the tumor showed robust fluorescent signal, suggesting that MMPF NPs were able to diffuse into the tumor even without a targeting ligand. Although the preclinical success of such a multimodal imaging platform addresses one major limitation of MPI—as it provides information only about NPs and their distribution, MPI by itself is unable to collect morphological data. The combination of MPI with CT or MRI would, therefore, allow for more complete anatomical visualization.

Additionally, it has been proposed that the underlying physics of MPI, which enable MNP imaging, may be simultaneously applied to enable real-time and non-invasive magnetic nanothermometry (MNT). This is significant given the current inability to accurately measure temperatures within the treatment region without inserting invasive thermal probes into the brain. There seems to be a consensus within the field that the future success of MHT relies on the creation of a single system capable of concurrent MHT-MPI-MNT. Notably, in 2023, Buchholz et al. designed such a system, with promising preliminary data showing the ability for combined hyperthermia, imaging, and thermometry in situ [50].

## 4. Discussion

MHT is a novel therapeutic modality that has the potential to safely and non-invasively target tumors, and can be applied to recurrent gliomas refractory to CT and RT. When compared to other heat-based treatment modalities, MHT offers many unique advantages. Most notably, it is possible to perform multiple sessions of non-invasive MHT due to the persistence of the MNPs around the delivery site for weeks to months, and the ability of the AMF to penetrate the skin and bone. This versatility allows for the unique coordination of treatment between MHT, CT and RT that has not been possible in the brain with current modalities (i.e., LITT). However, significant technological advancements are needed to achieve precise, real-time thermometry during MHT, as well as the real-time visualization of the MNPs within the brain. Imaging is crucial to ensure the correct thermal dosage and targeting.

While MNPs for MHT have been studied extensively in glioma cells in culture, their application in animal glioma models has been more limited. That said, there are currently nine studies which have evaluated the application of MNPs in various forms in brain tumor therapy. All demonstrate therapeutic promise in allowing for increased treatment efficacy and frequency while minimizing the potential adverse effects associated with current treatments, such as RT.

Each of these different MNPs offer their own advantages and disadvantages. Magnetosomes are biodegradable, a fact that minimizes long-term accumulation in brain tissue and, as a result, reduces heating efficacy on repeat treatment sessions. Chitosan-coated MNPs showed rapid tumor clearance with fewer sessions; however, they were associated with increased risk of high iron accumulation in the liver. Disc-shaped MNPs demonstrated the capacity for increased mechanical tumor lysis by introducing intratumor apoptosis with limited involvement of surrounding healthy brain tissue, although limited data exist on if and how they are cleared. Nanoclovers showed improved tumor and vasculature penetration and potential combined efficacy with CT; however, the BBB disruption associated with their usage could facilitate potential tumor recurrence. SPIONs have been studied in a more comprehensive evaluation of optimized therapeutic conditions and have shown significant potential for tumor mass reduction; however, further study on tumor regression following MHT and the associated survival benefit in animal glioma models is still needed. The differences between these nanoparticle formulation types warrant additional investigation as they may provide further opportunities for the individualized therapy of brain tumors.

Common across all nanoparticle formulations covered in this review is the importance of accurate, real-time nanoparticle visualization. To this end, the application of MPI in the neurooncology field has undergone significant advancement in recent years. In mouse brain tumor models, MPI has been used to detect micromolar concentrations of metal nanoparticle probes, following their uptake by brain tumor cells [51,52]. This may have promising implications in the field of neuro-oncology, as a major obstacle in brain tumor treatment planning is the imprecise identification of the tumor margins on MRI, which is the current gold-standard imaging modality for brain tumor patients. MPI has similarly shown promise in neurotrauma applications. In a mouse model of traumatic brain injury (TBI), MPI following NP administration enabled the determination of the location, depth, and severity of intracranial hematomas [52,53]. Furthermore, the NP signal detectable by MPI persisted for more than two weeks without significant intensity reduction. Taken together, these findings hold great promise for the application of MPI to MHT for brain tumors, in which the targeted tumor area may be located deep within the brain, and the treatment course consists of multiple sessions that may span several weeks.

With MHT emerging as a promising therapeutic option for gliomas—and solid tumors more broadly—evaluating the efficacy and safety of novel superparamagnetic MNPs is becoming progressively more important. In this regard, the European company, MagForce, has been the most prominent industry leader. A producer of both MNPs and AMF generators for human use [54,55], MagForce has been involved in several of the most recent clinical studies of MHT application in brain tumor patients. In their prospective, single-arm multicenter phase II clinical trial, 59 patients with recurrent or progressing GBM were recruited between 2005 and 2009 and underwent MHT, followed by monitoring at three-month intervals. Patients received intracranial instillation of MFL AS1, an aqueous suspension of superparamagnetic iron–oxide aminosilane-coated MNPs, with an iron concentration of 112 mg/mL [54]. The authors reported that the median overall survival from the diagnosis of the first tumor recurrence was 13.4 months, a substantial increase from the 6.2 months reported by Stupp et al. in their phase III study comparing combination RT and TMZ to RT alone [56].

It is important to note that MagForce filed for insolvency in 2022, and securing sufficient funding poses a major obstacle for the further evaluation of MHT application to brain tumor therapy. One reasonable path forward available is to leverage the sustained growth of nanomedicine. Since its inception, the United States National Nanotechnology Initiative (NNI) has consistently increased its annual budget, surpassing $2 billion in 2019 [57]. Concurrently, the global nanomedicine sector—valued at $170 billion in 2022—is projected to nearly triple in size over the next decade [53]. Furthermore, in their recent article, Fan et al. report that there are at least 15 cancer nanomedicines approved globally, with at least another 80 novel ones undergoing evaluation in more than 200 ongoing clinical trials [58]. Although MHT has shown significant promise in initial studies and offers unique benefits over other heat-based therapies for brain tumors, its application within the brain is still at its beginnings. The future clinical success of MHT relies on researchers continuing to refine MNP design as well as other factors involved in the therapy’s application (i.e., intracranial nanoparticle delivery techniques, AMF generator design, non-invasive real-time MNP thermometry and imaging). Taken together, this promising treatment—which has yet to reach its full potential—may be well-positioned for a major breakthrough in the coming years.

## Figures and Tables

**Figure 1 micromachines-15-00559-f001:**
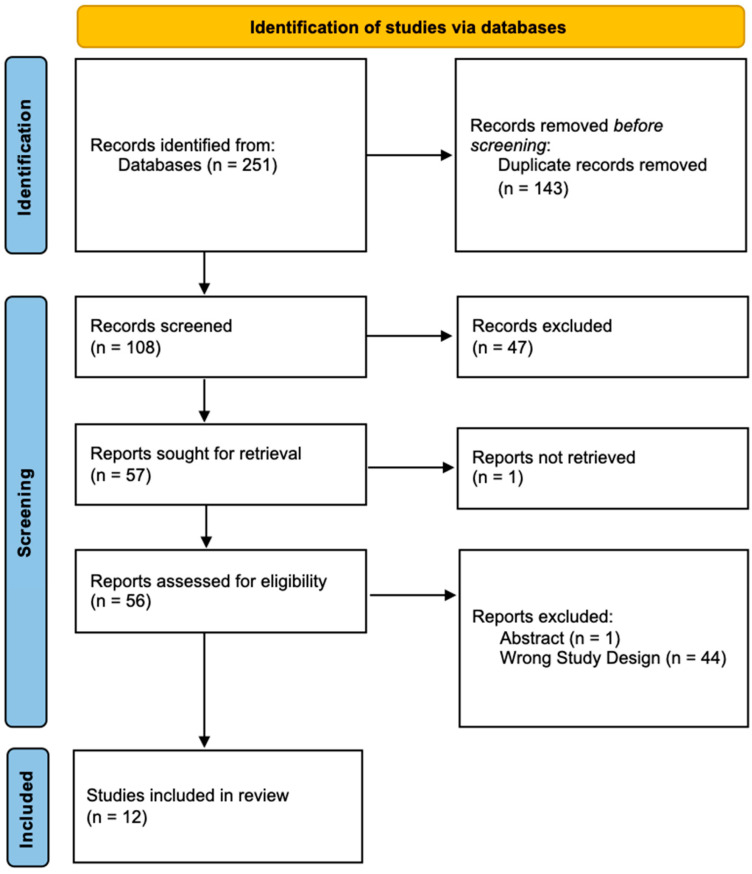
Prisma diagram. Prisma diagram outlining the selection of articles for this review.

**Figure 2 micromachines-15-00559-f002:**
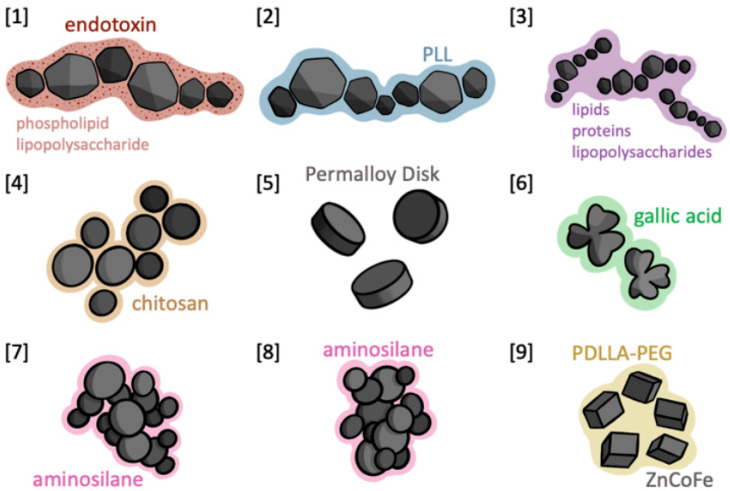
Nanoparticle Schematic. Illustrations of the 9 nanoparticles discussed in this study. Grey interior composed of Fe_3_O_4_ (with the exception of [9], as labeled) exterior composed of designated coating.

**Table 1 micromachines-15-00559-t001:** MHT Article Summary Data.

First Author	Year	Journal	Organism	Tumor Line	Nanoparticle
Alphandéry [37]	2017	Journal of Controlled Release	Mouse	U87-Luc	Magnetosomes
Alphandéry [38]	2017	Biomaterials	Mouse	U87-Luc	Magnetosomes coated with poly-l-lysine/iron-oxide nanoparticles (IONPs)
Alphandéry [39]	2019	Journal of Nanobiotechnology	Mouse	U87-Luc	Magnetosome chains
Chauhan [40]	2021	Biomaterials Science	Rat	C6	Chitosan-coated Fe_3_O_4_
Cheng [41]	2016	Journal of Controlled Release	Mouse	U87-Fluc-green fluorescent protein (GFP)	Disk-shaped permalloy magnetic particles
Liu [42]	2021	Nano Letters	Mouse	GL-261	Gallic acid-coated magnetic nanoclovers
Rego [43]	2019	Einstein (Sao Paolo)	Rat	C6	Aminosilane-coated superparamagnetic iron oxide nanoparticles (SPIONa)
Rego [44]	2020	International Journal of Molecular Sciences	Rat	C6	SPIONa
Wu [45]	2023	Journal of Controlled Release	Mouse	GL-261 and U87	Zinc- and cobalt-doped cubic IONPs

**Table 2 micromachines-15-00559-t002:** In Vivo Nanoparticle Parameters.

Particle	Mean Size	H_C	M_S	M_r/M_S	Mass (µg)	Volume (μL)	SAR (W/gFe)	Δt (°C)
CM [1]	~45 nm	~200–300 Oe	–	~0.35	40	2	4	~4
IONP [1]	17–20 nm	~120 Oe	–	~0.15	40	2	0	
M-PLL [2]	~45 nm core, 4–17 nm organic layer	~5 mT	–	~0.19	500–700	2	1.3	17.5
IONP [2]	17–20 nm	~11 mT	–	~0.15	500	2	0.2	8.5 °C
CM [3]	37.5 ± 5.2 nm (11 in vitro)	20 mT	–	0.3	40	2	4.7 ± 1.5	4 ± 1
Chitosan-coated Fe_3_O_4_ [4]	37 nm	–	71.5 emu/g	–	600–1000	–	460	7 (first), 9 (second)
Disk-shaped permalloy [5]	2 μm diameter	250 Oe	–	–	10	–	0.005	0
Gallic acid-coated magnetic nanoclovers [6]	20.7 nm	~700 Oe	~110 emu/g	–	25 mg/kg		~7000	To 48.4
SPIONs_Amin [7]	110 ± 5 nm	–	790.93 A/m	–	50	10	194.917	To 42
SPION_Amin [8]	100 nm	–	–	–	–	40 @ 4 coords	286	To 43
Zinc- and cobalt-doped cubic iron oxide nanoparticles [9]	52 nm	571 Oe	125 emu/g	–	50	–	3890	To 46

**Table 3 micromachines-15-00559-t003:** MHT Experimental Parameters.

Particle	Number of Animals	Age	Sex	Strain	MNP Injection PTI	Field Strength (mT)	Field Frequency (kHz)	Number of MS	AMF Duration (min)	Delivery Method	Survival
CM [1]	70 (10/group)	7 weeks	female	Nude mice ~20 g	D8	30	198	12–15	30	Intratumoral Injection	–
IONP [1]											
M-PLL [2]	54 (9/group)	5 weeks	female	Athymic nude mice ~18 g	D5	27	202	23–27	30	Intratumoral Injection	100% day 350 MSD > 350 (last mouse lived to 140 without heat vs. 50 for ctrl)
IONP [2]											MSD 57
CM [3]	60 (10/group)	6 weeks	female	Nude mice ~20 g	D8	27	198	3–15	30	Intratumoral Injection	50% survival at day 250 w/15 MS, 0 for 3 MS or ctrl MSD > 250
Chitosan-coated Fe_3_O_4_ [4]	18 (6/group)	10–12 weeks	male	Wistar Rats	D12–14	14 kA/m	335	2	20	Intratumoral Injection	Complete tumor inhibition in 32 days; no recurrence in 5 months post-mht
Disk-shaped permalloy magnetic particles [5]	18 (5/group, one group of 3)	6 weeks	male	Athymic nude mice 18–22 g	incubation or D3	1000	0.02	7	30	Incubation/Intratumoral Injection	MSD 63 treatment vs. 56 control
Gallic acid-coated magnetic nanoclovers [6]	70 (7/group)	5–6 weeks	female	C57BL6 (GL261 glioma), Balb/c (flank)	D14	27 ka/m	371	1	10	Tail vein/CED	43% survival at D60, 0% in control group after D42
SPIONs_Amin [7]	10	2 months	male	Wistar Rats 290–350 g	D22	200 G	874	1	40	Intratumoral Injection	–
SPION_Amin [8]	45 (9/group)	–	male	Wistar Rats 250–350 g	D14	300 G	309	1–3	30	Intratumoral Injection	–
Zinc- and cobalt-doped cubic iron oxide nanoparticles [9]	56 (7/group)	6–7 weeks	female	C57BL/6 and nude mice	D8	27 kA/m	410	–	10	Tail vein	Median survival of 60 days for experimental group

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
