# Peer review of "Recent Developments in Magnetic Hyperthermia Therapy (MHT) and Magnetic Particle Imaging (MPI) in the Brain Tumor Field: A Scoping Review and Meta-Analysis"

_micromachines, 2024, doi:10.3390/mi15050559_

Round 1
Reviewer 1 Report
Comments and Suggestions for Authors
Taking in consideration the review type of this article I would like to underline here some positive and some negative aspects regarding this review:
1. In terms of interest on the subject, brain tumors represent a very delicate issue in oncology, with little success rate on long term, so the idea of writing on this theme is positive. On the other hand, as the authors mentioned, it is very little literature on the subject (especially related to an increased rate of success) so in my opinion a review on this subject is too much.
2. The articles taken in consideration for this review are well selected, but as the rate of success is still very low and the results on animals not exactly relevant, the number of rats or mice reduced, I recommend:
a. to reorganize the Results part insisting shortly on both the benefits and mandatory on the disadvantages observed in each article because these can lead to essential results at the end
b. consider the paper from the perspective of a study (not a review) of what it is known until this moment
3. If the authors have personal experiences in the field, it should appear in the paper, as it gives higher credibility to the authors.
4. It should be highly underlined (including in the conclusion) the idea that from what it is known today, MNP’s are difficult to be delivered properly at the brain level and the methods MHT has potential, but it is still at its beginnings.
Author Response
We are writing to submit the revised version of our manuscript titled "Recent Developments in MHT and MPI in the Brain Tumor Field: A Scoping Review and Meta-Analysis," for consideration in Micromachines. We would like to express our sincere gratitude for the valuable comments and suggestions provided by the reviewers. These insights have been instrumental in enhancing the clarity, depth, and overall quality of our manuscript.
In response to the feedback received, we have carefully addressed each comment and made appropriate revisions to our manuscript. Our revisions aim to provide a more balanced and comprehensive analysis of the current advancements and challenges in the field of Magnetic Hyperthermia Therapy and Magnetic Particle Imaging for brain tumor treatment. We believe that these changes have significantly improved our manuscript, making it a valuable contribution to the journal and the broader scientific community interested in innovative treatments for brain tumors.
Reviewer 1:
- In terms of interest on the subject, brain tumors represent a very delicate issue in oncology, with little success rate on long term, so the idea of writing on this theme is positive. On the other hand, as the authors mentioned, it is very little literature on the subject (especially related to an increased rate of success) so in my opinion a review on this subject is too much.
Thank you for the valuable perspective. While we agree that there are few treatment options currently for aggressive forms of brain cancer, we view this as a compelling reason for why promising novel therapies should be highlighted and shared with the neuro-oncological community, as we aim to do in this review. Learning from the successes and limitations of these therapies is crucial for guiding the future direction of neuro-oncological research. Additionally, although MHT is relatively new as a treatment for brain tumors, using heat to treat brain tumors is well-established. Furthermore, MHT has been studied extensively as a treatment for cancer in other parts of the body. As such, we see this as an excellent time to summarize the current progress of MHT for the treatment of brain tumors, as the field is at a critical point in its development. Additionally, this review aims only to focus on some of the different types of nanoparticles used for intracranial MHT, it does not aim to summarize all literature on intracranial MHT.
- The articles taken in consideration for this review are well selected, but as the rate of success is still very low and the results on animals not exactly relevant, the number of rats or mice reduced, I recommend:
- to reorganize the Results part insisting shortly on both the benefits and mandatory on the disadvantages observed in each article because these can lead to essential results at the end
- consider the paper from the perspective of a study (not a review) of what it is known until this moment
Thank you very much for your thoughtful comments and constructive feedback regarding our manuscript. We appreciate the time you took to carefully review our work and the valuable suggestions you provided. In response to your recommendations, we have taken the opportunity to reorganize the Results section of our paper. Specifically, we have now included a more balanced discussion for each article reviewed, highlighting not only the benefits but also emphasizing the limitations and disadvantages observed. We agree with you that acknowledging these limitations is crucial for providing a comprehensive overview and can lead to essential insights for future research in this field.
- If the authors have personal experiences in the field, it should appear in the paper, as it gives higher credibility to the authors.
We would like to clarify that we are primarily composed of medical students who are passionate about advancing our understanding of magnetic hyperthermia therapy and magnetic particle imaging in the context of brain tumor treatment. While we are earnestly committed to contributing to this field through rigorous academic research, our practical experience, especially in conducting hands-on clinical or laboratory work related to MHT and MPI, is currently limited.
- It should be highly underlined (including in the conclusion) the idea that from what it is known today, MNP’s are difficult to be delivered properly at the brain level and the methods MHT has potential, but it is still at its beginnings.
In response to your suggestion to emphasize the challenges associated with MNP delivery in the brain and the nascent state of MHT, we have made significant revisions to our manuscript. Specifically, we have included a statement in our conclusion to highlight that, while MHT holds considerable potential for the treatment of brain tumors, it is still in the early stages of development. We underscored that the effective delivery of MNPs to the brain remains a critical challenge, and that the overall clinical success of MHT will depend on the continued refinement of multiple aspects of the therapy. These aspects include the design of MNPs, the development of advanced intracranial nanoparticle delivery techniques, improvements in AMF generator design, and the advancement of non-invasive real-time MNP thermometry and imaging technologies.
Reviewer 2 Report
Comments and Suggestions for Authors
This paper analyzes 251 articles and summarizes 12 articles analyzing magnetic heat therapy. Preliminary result show that MHT is an exciting treatment modality with unique advantages over current heat-based therapies for brain cancer. This study aims to summarize the development and application of magnetic heat therapy in recent times and suggest technological advances needed for MHT. The following modifications have been suggested.
1. Figure 2 seems to be like the screenshot. Its resolution is suggested to be improved.
2. In the manuscript, only 12 studies have been selected as the examples. Compared to photothermal therapy and sonodynamic therapy, unfortunately, there is no sufficient data to prove that MHT has enough benefits to be supported as the powerful tool in the therapy of brain tumors. In addition, more reference papers can be added in the revised manuscript.
3. It is recommended that Table 2 and Table 3 be made into graphs in order to visualize and clearly display the results.
4. More figures are suggested to be added in the revised manuscript.
Comments on the Quality of English LanguageExtensive editing of English language required
Author Response
We are writing to submit the revised version of our manuscript titled "Recent Developments in MHT and MPI in the Brain Tumor Field: A Scoping Review and Meta-Analysis," for consideration in Micromachines. We would like to express our sincere gratitude for the valuable comments and suggestions provided by the reviewers. These insights have been instrumental in enhancing the clarity, depth, and overall quality of our manuscript.
In response to the feedback received, we have carefully addressed each comment and made appropriate revisions to our manuscript. Our revisions aim to provide a more balanced and comprehensive analysis of the current advancements and challenges in the field of Magnetic Hyperthermia Therapy and Magnetic Particle Imaging for brain tumor treatment. We believe that these changes have significantly improved our manuscript, making it a valuable contribution to the journal and the broader scientific community interested in innovative treatments for brain tumors.
Reviewer 2:
This paper analyzes 251 articles and summarizes 12 articles analyzing magnetic heat therapy. Preliminary result show that MHT is an exciting treatment modality with unique advantages over current heat-based therapies for brain cancer. This study aims to summarize the development and application of magnetic heat therapy in recent times and suggest technological advances needed for MHT. The following modifications have been suggested
- Figure 2 seems to be like the screenshot. Its resolution is suggested to be improved.
We have included a separate pdf of this figure with a higher resolution.
- In the manuscript, only 12 studies have been selected as the examples. Compared to photothermal therapy and sonodynamic therapy, unfortunately, there is no sufficient data to prove that MHT has enough benefits to be supported as the powerful tool in the therapy of brain tumors. In addition, more reference papers can be added in the revised manuscript.
We agree that currently there is not enough evidence to support the immediate inclusion of MHT as a treatment option for brain tumors. The therapy is still in its development, and the goal of this review is not to convince the reader why MHT should be currently implemented clinically, but rather to inform the reader of the current state of the field and highlight why further research is warranted given promising preliminary results.
- It is recommended that Table 2 and Table 3 be made into graphs in order to visualize and clearly display the results.
After careful deliberation, we respectfully wish to convey our perspective that these specific tables are designed to lay out the parameters and characteristics of individual nanoparticles rather than to present direct results or findings from experimental data. The structured format of the tables is intended to provide a clear, concise, and comparative overview of the various attributes and experimental conditions associated with each type of magnetic nanoparticle discussed in our review. This format allows readers to easily reference and compare the specific properties and conditions relevant to the development and application of MHT and MPI in the brain tumor field.
- More figures are suggested to be added in the revised manuscript.
The process of creating additional figures that meet the high standards of quality and accuracy we aim for requires a significant amount of time, both in terms of conceptualization and execution. Given the tight turnaround time required to submit our revisions, we are concerned that rushing this process may compromise the quality of our work and fail to do justice to the valuable data and insights we wish to convey. Therefore, while we acknowledge the potential benefits of adding more figures to enhance the manuscript, we respectfully request to proceed with the submission without these additional visual elements at this time. We believe that the current text and existing figures adequately support the narrative and findings of our study.
Round 2
Reviewer 1 Report
Comments and Suggestions for Authors
I appreciate the authors took in consideration my previous suggestion. A balance between benefit of the method and its limitation was strongly required. I decided to recommend for publication the manuscript, but hoping that in the near future, new and consistent results to be investigated.